# Depression and Suicide Ideation among Suicide-Loss Survivors: A Six-Year Longitudinal Study

**DOI:** 10.3390/ijerph192416561

**Published:** 2022-12-09

**Authors:** Yossi Levi-Belz, Shai Birnbaum

**Affiliations:** 1The Lior Tsfaty Center for Suicide and Mental Pain Studies, Ruppin Academic Center, Emek Hefer 4025000, Israel; 2Clinical Psychology M.A. Program, Ruppin Academic Center, Emek Hefer 4025000, Israel

**Keywords:** suicide, depression, suicidal ideation, belongingness, bereavement

## Abstract

Suicide is not only a tragic end of life but also may be the beginning of a very challenging life for those left behind. Suicide-loss survivors (SLSs) are individuals who were exposed to the suicide of a close family member or a friend and endure highly emotional distress. The psychological stance and reactions of SLS are deeply colored by painful, intense emotions that are expressed through different psychiatric symptoms, including depression and suicide ideation (SI). The present study investigated the long-term effects of interpersonal factors such as social support, self-disclosure, thwarted belongingness, and perceived burdensomeness on depression and suicidal ideation among SLS. One hundred fifty-two Israeli SLS, aged 20–72, participated in this longitudinal study, during which their suicide ideation and depression levels were assessed at four points over six years (T1-baseline, and two (T2), four (T3), and six (T4) years after baseline). At the last time point, interpersonal factors were also assessed. SLSs’ interpersonal variables significantly predicted depression and SI levels beyond their usual trajectories over the years. Significant correlations were found between both perceived burdensomeness and thwarted belongingness and depression levels at all measurement points. Moreover, thwarted belongingness was found to be a significant moderator of the relationship between former and current depression levels, as the contribution of depression-T3 to depression-T4 was lower among SLSs with low thwarted belongingness (b = 0.14, CI = 0.05–0.34) compared with SLSs with high thwarted belongingness (b = 0.25, CI = 0.22–0.45). These findings emphasize the vital healing role of interpersonal factors such as belongingness, as they may attenuate depression and SI symptoms over time. Hence, clinicians should focus on therapies that boost interpersonal interactions and belongingness, as they seem to be crucial stepping stones on the way to recovery. Moreover, national programs should be implemented to offer SLSs targeted interventions to reduce distress and depression in the aftermath of suicide loss.

## 1. Introduction

Suicide is not only a tragic end of life but may also be the beginning of a very challenging life for those left behind. Contemporary estimates highlight the profound impact of each suicide on about 60 suicide-loss survivors (SLSs), referring to individuals exposed to the suicide, including family members, friends, co-workers, classmates, or therapists [1,2,3,4]. Thus, between 48 million to 500 million people could be considered SLSs annually [5]. The psychological stance and reactions to suicide loss are deeply colored by painfully intense emotions like shame and guilt, which differ at least partly from other bereaved individuals [6,7,8]. Accordingly, research has highlighted that SLSs are characterized by acute mental pain expressed through highly significant psychiatric symptoms [9]. Serious adverse health, such as higher levels of grief complications and, importantly, higher levels of depression [2] and suicidal ideation [10,11], has been found to be associated with SLSs. These deleterious emotional effects of suicide highlight the importance of studying the influence of suicide on family and friends, as well as ways we can ease the pain and agony that accompany suicide loss.

Only few studies have examined trajectories of depression and SI among SLSs over time. Kõlves et al. [12] examined mental health reactions among SLSs and individuals bereaved by other sudden death events six, 12, and 24 months after their loss, finding that depression levels were significantly reduced over time. In a recent longitudinal study, Levi-Belz and Aisenberg [10] found SI and complicated grief to accompany SLSs, directly and indirectly, over a period of almost 4 years. Another study established a connection between SLSs and subsequent long-term depression 8–10 years following the suicide, finding that the risk of depression and SI decreased over time [13]. While these findings are meaningful, many more studies are needed to explain the contribution of psychological factors to SLSs’ depression and SI over time [11]. This study aimed to examine the possible moderators in depression and SI trajectories among SLSs in a six-year longitudinal study.

Several studies have already highlighted factors that might moderate depression and SI consequences among SLSs. For example, De Groot et al. [13] found that children who have lost a parent and individuals with a mental health history were at increased risk 8–10 years after the suicide loss. Other studies found that religious beliefs contribute to psychological adjustment following bereavement [14]. Some self-traits, such as a low sense of control in life, low self-efficacy, and high neuroticism, accounted for elevated depression levels [13]. More recently, another self-attribute, self-forgiveness, was found to exhibit a unique protective effect for SLSs. Self-forgiveness was found to be related to both lower depression and SI for SLSs and had a greater ameliorating effect in this population than in other bereaved individuals [15]. Guilt was also associated with depression and moderated the association between time since the suicide and depressive symptoms [16].

Interestingly, interpersonal factors have gained considerable attention as moderators of depression and SI among SLSs. Much of the spotlight has been centered on perceived social support levels, defined as “an exchange of resources between at least two individuals perceived by the provider or recipient to be intended to enhance the well-being of the recipient” [17]. Investigations have indicated that for SLSs, perceived social support manifests a reverse relationship with depression and SI symptoms [18,19]. In a cross-sectional study [20] relying on the data of 195 convenience-sample SLSs, higher levels of perceived social support were significantly related to lower levels of both depression symptoms and SI. As the study concluded, social support may play an important role in suicide postvention and should be further investigated to better understand the mechanism of its effect [20].

A construct related to perceived social support is self-disclosure, referring to the process by which individuals let themselves be known by others [21]. Self-disclosure was found to be an important protective factor against different psychopathologies among various samples [22,23]. For SLSs, self-disclosure was found to lead to reduced levels of complicated grief, even beyond the natural trajectory of complicated grief over time [24]. In another study, self-disclosure was found to be a protective factor against grief difficulties [25] among a sample of mostly SLSs (131 out of 147 bereaved). However, the protective role of social support and self-disclosure in SLSs’ depression and SI remains unclear since longitudinal studies on the moderating role of social support on mental health characteristics among SLSs are scarce.

Interestingly, the relationships between interpersonal factors and mental difficulties such as SI have primarily been examined through Thomas Joiner’s interpersonal theory of suicide [26]. This theory emphasizes two main dimensions that may influence depression and SI: perceived burdensomeness (PB) and thwarted belongingness (TB; [27]). PB reflects an affective-cognitive state reflecting the view that one’s existence is a burden to friends, family members, or society. It comprises self-hate (e.g., “I hate myself”) and feelings of liability (e.g., “I make things worse for people in my life”). TB describes a distinguished cognitive-affective state portrayed by the painful feeling of being alienated or external to one’s family, friends, and other valued groups [27]. Since these dimensions have been recognized as comprising the heart of interpersonal interaction [26], it can be suggested that they may contribute an important aspect of interpersonal qualities, thus moderating SLSs’ depression and SI.

Recent studies have noted that lower TB levels may serve as a protective factor against SLSs’ complicated grief and depression [28]. Levi-Belz and Aisenberg [10] conducted a four-year longitudinal design study and found that TB, but not PB, increases SI for SLSs. On the other hand, PB was found to increase depression levels in various populations [29,30]. However, important information concerning the possible moderation of PB and TB on the evolvement of depressive symptoms over time since suicide loss has yet to be examined.

### The Present Study

In the present study, we aimed to explore to what extent interpersonal variables may help SLSs cope more effectively with the deleterious consequences of suicide loss, using 6 years of longitudinal data with four measurement points (T1-index measurement, T2-two years after T1, T3-four years after T1, T4-six years after T1). Specifically, we intended to broaden knowledge regarding depression and SI levels of SLSs over time and examine the moderating effect of interpersonal variables (i.e., perceived social support, self-disclosure, TB, and PB) on SLSs’ depression and SI trajectories over time. To date, only a few studies have investigated the contribution of interpersonal factors to the longitudinal course of depression and SI levels among SLSs.

We posited the following hypotheses:Interpersonal variables of social support, self-disclosure, TB, and PB at T4 will contribute to depression and SI symptoms beyond the effect of previous depression and SI levels at T1–T3.Interpersonal variables will moderate the relationship between previous depression and SI levels and current depression and SI levels (T4).

## 2. Method

### 2.1. Participants

Participants were 189 Israeli SLSs who were assessed at the T1 measurement point. Of these, 156 (82.5%) were assessed at T2 (1.5 years after T1), and 152 (80.4%) were assessed at T3 and T4 (3.5 years and 5.5 years after T1, respectively). The participants were recruited through social media groups of SLSs in Israel, primarily through the national agency for SLSs in Israel (“The Path to Life”).

Of the 37 participants who dropped out from the measurement after T1, 23 could not be located, 13 did not respond to the T2 invitation letter, and one died (due to cancer). We found no significant demographic or psychological differences between participants who completed all four measurements and those who completed only T1. Thus, the current study comprises 152 SLSs (130 females) aged 20–72 at T1. Participants eligible for the study included those who had lost a family member or another close friend due to suicide [2]. Exclusion criteria were the inability to read and write in Hebrew and being under 15 at the time of the suicide (thus including only SLSs whose suicide loss occurred when they were adolescents or older).

### 2.2. Procedure

Potential participants were required to affirm their willingness to participate by signing an informed consent form. They were also informed of the risks and compensation procedures and were assured anonymity, confidentiality, and their right to withdraw from the study at any time. Those eligible for the study then completed the online questionnaire in Hebrew (using Qualtrics online survey software). At the end of the T1 measurement point, participants were asked if they would agree to be approached again. Those who agreed were invited again at subsequent measurement points (with a gap of approximately two years between measurements). All participants were compensated for their participation (gift vouchers of US $50 were granted for each measurement). The study was approved by the ethics committee at the Ruppin Academic Center.

### 2.3. Measures

We used perceived burdensomeness, thwarted belongingness, social support, and self-disclosure as independent variables in this study. We used depression and suicide ideation levels as dependent variables. The following section describes each variable and its measurement.

#### 2.3.1. Perceived Burdensomeness (PB) and Thwarted Belongingness (TB)

PB and TB were assessed by the Interpersonal Needs Questionnaire (INQ; Ref. [31]), a 10-statement inventory used to assess either TB (e.g., “These days, other people care about me” [reverse-scored]) and PB (e.g., “These days, I feel like a burden on the people in my life”), with five items presented for each subscale. Each item is presented on a 7-point Likert scale ranging from 1 (*Not at all true for me*) to 7 (*Very true for me*). Higher scores reflected greater TB and PB. TB and PB were measured only at T4. In this study, we used the Hebrew translation of the INQ, which was used in various studies (e.g., Ref. [32]). The internal consistency for the current sample for PB was α = 0.90, and for TB, α = 0.85.

#### 2.3.2. Social Support

The Multidimensional Scale of Perceived Social Support (MSPPS) was used to assess social support ([33]). The MSPPS is a 12-item questionnaire measuring the perceived adequacy of social support from three sources: family members, friends, and significant others. The 12 items were rated on a 7-point Likert-type scale, ranging from 1 (*disagree very strongly*) to 7 (*agree very strongly*). Higher summed scores indicate greater levels of perceived social support. The MSPSS has good internal and test–retest reliability and a fairly stable factorial structure [34]. It has been used widely in many languages, including Hebrew [35,36]. Social support was measured at T4. The internal consistency for the current sample was α = 0.92.

#### 2.3.3. Self-Disclosure

The Distress Disclosure Index (DDI; Ref. [37]) was used to assess the distress disclosure and negative emotions levels. The DDI measures the inclination to disclose distressing information, thoughts, personal problems, and unpleasant emotions across time and situations. The 12 items of the DDI were presented on a 5-point Likert scale, ranging from 1 (*strongly disagree*) to 5 (*strongly agree*). Higher DDI scores reflect a higher degree of disclosing distress to others. The DDI is a highly reliable and valid measure, introducing high coefficients across different samples [38]. Self-disclosure was measured at T4. The internal consistency for the current sample was α = 0.94.

#### 2.3.4. Depression

PHQ-9 is a widely used self-administered measure of depression, comprising nine items that reflect the nine DSM-V diagnostic criteria for major depression [39]. Each item assesses the frequency of that symptom over the past two weeks, rated on a 4-point ordinal scale: (0) *Not at all*, (1) *Several days*, (2) *More than half the days*, (3) *Nearly every day*. The PHQ-9 is a valid and reliable measure of depression [39]. The PHQ-9 was validated against professional diagnoses of MDD, resulting in 88% sensitivity and 88% specificity. Depression was measured at three time points (it was not measured at T1) and was found to have adequate to high reliability at T2 (α = 0.89), T3 (α = 0.92), and T4 (α = 0.90).

#### 2.3.5. Suicide Ideation (SI)

Due to the longitudinal nature of the study, only a single item tapping the frequency of current SI was assessed (“How often have you thought about killing yourself in the past year?”), the second item from the four-item Suicidal Behaviors Questionnaire-Revised (SBQ-R; Ref. [40]). Several studies have used this item to assess suicidality [10,41], and there is strong evidence for a single item’s predictive ability and relevance in suicidality assessment [42,43]. The SI item is presented on a 5-point Likert-type scale, ranging from 1 (*never*) to 5 (*very often*; five or more times). Higher scores indicate increased levels of suicide risk. In this study, we administered the Hebrew translation of the SBQ, which has been used in various studies (e.g., Ref. [44]). SI was assessed at all four measurement points.

#### 2.3.6. Demographic and Suicide-Related Characteristics

In addition to the above measures, demographic and suicide loss characteristics were collected for each participant, including the age, gender, family status, the ages of the SLSs and the deceased at the time of the suicide, the time since the suicide event, and the participants’ relatedness to the person who died by suicide.

### 2.4. Data Analysis

First, we performed a series of Pearson correlation tests and ANOVA analyses with Bonferroni correction to examine the relationships between the study variables. Then, we conducted hierarchical multiple regression analyses with depression-T4 and SI-T4 as dependent variables to further investigate the effect of interpersonal variables and hypothesized interactions beyond the effect of previous depression or SI levels. Lastly, as recommended by Aiken et al. [45], all continuous predictor variables were standardized, as were the cross-product interaction terms. To examine the nature of the interaction within a regression framework, moderation analysis was performed using the PROCESS macro (Model 1; Ref. [46]). All analyses were conducted using the Statistical Package for the Social Sciences (SPSS, v26.0 for Windows, Armonk, NY, USA). The significance level for all statistical tests was set at 0.05.

## 3. Results

### 3.1. Demographic Information of the Sample

At T4, the mean age of the sample was 45.9 (*SD* = 14.7). Regarding the participants’ family status, 72 (47%) were married, 61 (40%) were single, nine were divorced (6%), and 10 (7%) were widowed. Most participants reported their Jewish religiosity as secular (115, 75%), with a minority (28, 18%) reporting to be religiously observant. Regarding socioeconomic status (SES), 36 (23.8%) participants reported their SES as very low, 39 (26%) as low, 38 (25%) as medium, and 38 (24%) as high. Regarding schooling, almost all participants (*n* = 151, 99.3%) reported completing at least 12 years, and almost 70% (*n* = 105) reported having a college degree. Regarding residential areas within Israel, 55% of the sample reside in central Israel, 20% in the north, and 25% in the south. In all, the participants reported 40 different cities, with the largest numbers in Tel Aviv (15%) and Haifa (7%).

### 3.2. Suicide-Related Characteristics

The participants reported various levels of relationship to the deceased: 29 were parents to the deceased (18.6%), 26 children (16.7%), 43 siblings (27.6%), 16 spouses (10.3%), 13 (8.4%) other family relatives, and 29 (18.6%) best friends. At T1, time since the suicide varied among the participants (*M*_months_ = 80), with a range of 6 to 200 months: 27 participants (17.7%) had lost their significant other within 24 months prior to T1, 38 (25%) within 24–48 months, 44 (29%) within 48–72 months, and the remainder (43; 28.2%) six years or more prior to T1. At the time of the suicide, the participants’ mean age was 31.1 (*SD* = 15.3), ranging from 16 to 62. All participants reported being devastated by the suicide, ranging between extremely devastated (47, 30.8%), highly devastated (84, 55.8%), and devastated (20, 13.4%).

### 3.3. Depression and Suicide Ideation over Time

To examine SLSs’ depression and SI trajectories, two repeated measure ANOVA analyses were conducted (see Figure 1). For depression, the effect of time was found to be insignificant. For SI, the effect of time was significant, *F*(3, 453) = 75.02, *p* < 0.001, η2 = 0.33]. SI levels increased from T1 (*M* = 1.74, *SD* = 1.15) to T2 (*M* = 2.85, *SD* = 1.05) and then decreased at T3 (*M* = 2.38, *SD* = 1.05) and at T4 (*M* = 1.54, *SD* = 0.59).

### 3.4. Relationships between the Study Variables

Pearson correlations were calculated between the study variables to examine associations between interpersonal factors and depression/SI over time. As can be seen in Table 1, social support was negatively correlated with depression-T2 (*r*_150_ = −0.37, *p* < 0.001), but not with any other measures of depression or SI. Self-disclosure was negatively associated with depression-T3 and depression-T4, but not with depression-T2 nor with SI levels. PB-T4 was associated with depression at all time measures: PB-T4 was associated with depression-T2 (*r*_150_ = 0.32, *p* < 0.001), depression-T3 (r_150_ = 0.26, *p* = 0.002), and depression-T4 (*r*_150_ = 0.56, *p* < 0.001). PB was also associated with SI-T1 (*r*_150_ = 0.26, *p* = 0.001) and SI-T4 (*r*_150_ = 0.31, *p* < 0.001). Similarly, TB was associated with depression-T2 (*r*_150_ = 0.19, *p* = 0.02), depression-T3 (*r*_150_ = 0.2, *p* = 0.016) and depression-T4 (*r*_150_ = 0.42, *p* < 0.001). TB was also associated with SI-T1, (*r*_150_ = 0.25, *p* = 0.002) and SI-T4 (*r*_150_ = 0.32, *p* < 0.001).

### 3.5. Effect of Interpersonal Variables on Depression at T4

To examine the contribution of interpersonal factors on depression over time, a hierarchical regression analysis was applied, with depression-T4 as the dependent variable (see Table 2). To control statistically for the time since suicide, this variable was entered into the equation in the first step. In the second step, the main effects of depression-T2 and depression-T3 were entered into the equation. Social support-T4 and self-disclosure-T4 were entered in the third step. In the final step, PB-T4 and TB-T4 were entered into the equation. This analysis enabled us to interpret the effect of interpersonal variables beyond the influence of time since suicide and previous depression levels.

Overall, the total set of variables explained 49.3% of the variance for depression-T4, *F*(7, 144) = 19.98, *p* < 0.001. As seen in Table 2, time since suicide was not found to be a significant predictor. In Step 2, a model incorporating the main effect of depression-T2 [Beta = 0.32, *t*(148) = 4.36, *p* < 0.001] and depression-T3 [Beta = 0.36, *t*(148) = 4.94, *p* < 0.001] accounted for 30% of the variance and significantly predicted depression-T4, *F*(2, 148) = 31.67, *p* < 0.001. In Step 3, the main effect of social support [Beta = −0.19, *t*(146) = −2.29, *p* = 0.024] and self-disclosure [Beta = −0.20, *t*(146) = −2.56, *p* < 0.012] significantly predicted depression-T4 and accounted for another 3.6% of the total variance, *F*(2, 146) = 3.99, *p* = 0.021. In the final step, PB-T4 [Beta = 0.27, *t*(144) = 2.96, *p* = 0.004] and TB [Beta = 0.22, *t*(144) = 2.17, *p* = 0.032] significantly and positively contributed to depression-T4 beyond all other variables, accounting for another 15.7% of the total variance, *F*(2, 144) = 22.23, *p* < 0.001].

To understand the direct effect of each one of the interpersonal variables, we performed an additional regression analysis with all of the interpersonal variables as predictors of depression at T4. Social support (Beta = −0.16, *p* = 0.042; **∆**R^2^ = 0.02 *F* change = 3.55, *p* = 0.042), PB (Beta = 0.42, *p* = 0.000; **∆**R^2^ = 0.15 *F* change = 40.06, *p* = 0.001) and TB (Beta = 0.27, *p* = 0.007; **∆**R^2^ = 0.05 *F* change = 8.57, *p* = 0.007) were all found to be significant predictors of depression-T4, whereas self-disclosure contributed negatively but insignificantly to depression-T4.

### 3.6. Interpersonal Effect on Suicide Ideation at T4

To determine whether PB and TB can predict SI for SLSs, a hierarchical regression analysis was applied. A regression equation was constructed with SI-T4 as the dependent variable (see Table 3). To control for the time since suicide, the variable was entered into the equation in Step 1. In Step 2, the main effects of SI-T1, SI-T2, and SI-T3 were entered into the equation. Social support-T4 and self-disclosure-T4 were entered in Step 3. In the final step, PB-T4 and TB-T4 were entered into the equation. This analysis enabled us to interpret the effect of interpersonal variables beyond the influence of time since suicide and previous SI levels.

Overall, the total set of variables explained 30.5% of the variance for SI-T4, *F*(8, 143) = 7.847, *p* < 0.001. In Step 1 (see Table 3), time since suicide was not found to be a significant predictor. In Step 2, a model incorporating the main effect of SI-T1, SI-T2, and SI-T3, accounted for a further 20.2% of the variance in predicting SI-T4, *F*(3, 147) = 12.763, *p* < 0.001. Both SI-T1 [Beta = 0.30, *t*(147) = 4.06, *p* < 0.001] and SI-T3 levels [Beta = 0.25, *t*(147) = 3.30, *p* < 0.001] were highly correlated with SI-T4. In Step 3, the main effect of social support-T4 [Beta = −0.19, *t*(145) = −2.42, *p* = 0.017] significantly predicted lower SI-T4, but together with self-disclosure did not add a significant explanation to the SI-T4 variance. In the final step, PB-T4 and TB-T4 together accounted for 5.8% in predicting SI-T4 after all other variables had been entered, *F*(2, 143) = 5.955, *p* < 0.003.

To understand the direct effect of each interpersonal variable discretely, we performed an additional regression analysis with all the interpersonal variables as predictors of SI at T4. Social support (Beta = −0.18, *p* = 0.044; **∆**R^2^ = 0.025 *F* change = 4.88, *p* = 0.030) and TB (Beta = 0.19, *p* = 0.050; **∆**R^2^ = 0.02 *F* change = 3.56, *p* = 0.007) were found to be significant predictors of SI-T4, whereas self-disclosure and PB were not related significantly to SI-T4.

### 3.7. Moderation Analysis

Following the hierarchical regression results, we employed moderation analyses of significant interactions using the PROCESS macro (Model 1; Ref. [46]). Moderation analyses were conducted with depression-T4 as the dependent variable and depression-T3 as the independent variable. TB levels served as the moderator. The trajectory of depression (depression-T2) was entered as a covariate. As seen in Figure 2, a significant interaction was found between depression-T3 and TB in predicting depression-T4, b = −0.02, SE = 0.01, 95% CI [−0.03, −0.01], *t*(147) = 1.95, *p* = 0.05. Probing the interaction revealed that for SLSs with low/moderate/high levels of TB, depression-T3 positively contributed to depression-T4: the correlation between depression-T3 and depression-T4 was lower for low TB: b = 14.9, SE = 0.07, 95% CI [0.05, 0.34], *t*(147) = 2.71, *p* = 0.001; higher for moderate *SD*: b = 20.27, SE = 0.05, 95% CI [0.15, 0.38], *t*(147) = 4.76, *p* < 0.001, and the highest for high TB: b = 25.65, SE = 0.06, 95% CI [0.22, 0.45], *t*(147) = 5.65, *p* < 0.001.

### 3.8. DAG Analysis

To establish the correctness of the presented interaction in Figure 1, we conducted a directed acyclic graph (DAG) of an alternative moderation between the variables, in which the independent and the outcome measures switch. As seen in Figure 3, the DAG presentation shows no interaction between depression-T4 and TB when predicting depression-T3 as outcome measures (b = 0.00, SE = 0.01, 95% CI [−0.03, 0.03], *t*(147) = 0.21, *p* = 0.79). These findings help confirm the above interaction.

## 4. Discussion

This study aimed to investigate the contribution of interpersonal factors to SLSs’ course of depression and suicide ideation over time. To our knowledge, this study is the first to address the moderating role of interpersonal factors in decreasing depression and SI beyond their natural trajectories among SLSs. As expected, significant trajectories were found regarding depression and SI over time, meaning that both depression and SI in T1-T3 were strongly correlated with depression and SI at T4. However, importantly as hypothesized, the interpersonal factors of social support, self-disclosure, PB, and TB contributed to depression and SI at T4 above and beyond their natural trajectories. Thus, SLSs with higher levels of interpersonal abilities showed lower levels of depression than those with lower levels of interpersonal abilities.

Furthermore, TB was found to be a significant moderator in the link between depression at T3 and T4, as SLSs who reported a higher belongingness experience showed lower levels of depression at T4 and a lower association between depression at T3 and T4, compared with SLSs who reported moderate or low levels of a belongingness experience. Thus, it may be suggested that belongingness serves as a buffer for depression among SLSs. Taken together, these results highlight the critical role of interpersonal factors, specifically the belongingness experience, as a possible protective factor against developing depression in the aftermath of suicide loss.

What can explain the importance of belongingness and social support as a protective factor against depression among SLSs? Several explanations warrant mentioning. First, when SLSs feel thwarted belongingness and lack of social support, it may reflect a lower quality of social relationships with their surroundings. Such relationships may amplify the oppressive experience of blame and shame, which characterize many SLSs (e.g., Refs. [5,47]) and hinder their recovery from depressive symptoms. The more acute grieving process of SLSs with high TB [10] may also play a role in the association between low social relationships and depression, promoting lasting symptoms of depression for the high TB group. On the other hand, a higher sense of belongingness, viewed as one of the fundamental human psychological needs [48], may function as a buffer against the stigma and shame experienced by SLSs [5,47] and, as a result, shield SLSs from mental pain and even depression.

From another perspective, it can be suggested that the belongingness experience and better social relationships may facilitate more effective coping with depression for SLSs. Moreover, a greater sense of belongingness and social support may lead to greater prospects of self-disclosure to others, which, in turn, can promote sharing of intimate thoughts and emotions and create closer, more intimate, supportive bonds and feeling loved [24]. Together, these factors may help reduce loneliness, one of the primary characteristics of SLSs [49], closely associated with depression levels [50].

Low TB levels have also been shown to be enhanced by engaging in interpersonal activities such as disclosing intimate information and gaining support from the SLSs’ surroundings [51]; both reflect feelings of being connected to and supported by significant others. It can be suggested that sharing personal information with others (reflecting a sense of belongingness) and revealing and processing emotional aspects of the trauma may facilitate intimacy and togetherness among SLSs, which are recognized as protective factors against distress. The knowledge that the individual is not bound to face loneliness and rejection from others may help alleviate the devastating emotional aspects of the suicide loss, such as depression [52]. This notion aligns with several studies highlighting the impact of social factors on depression reduction (e.g., [53,54]).

Our study had some methodological limitations. First, we used self-report questionnaires in this study, which may be less reliable and partially biased to self-presentation and inaccurate reporting of mental health-related items. Future studies should apply other types of measures. Second, most participants were members of either an SLS nonprofit organization or the Internet forum dedicated to dealing with suicide loss. Participants in these mutual help forums may be inclined to seek more support and be less isolated than other SLSs, making the current sample less representative. The relatively low levels of depression that the participants reported may be evidence of the low representativeness of the general SLS population, which is known to suffer from high distress following the suicide event. Lastly, interpersonal factors were measured in our study only at T4. Thus, these factors may have been influenced by the levels of depression and SI at T1-T3. Future studies should seek to examine these factors before the suicide loss as well, shedding light on pre-trauma characteristics that may help SLSs cope with depression and SI following their loss.

## 5. Conclusions and Implications

Taken together, these findings highlight the important role of interpersonal factors in decreasing SLSs’ depression and SI levels. It may be suggested that when an SLS feels supported by others, senses that they belong to family and community, and can communicate pain without feeling like a burden, their healing process accelerates beyond the natural depression and SI course. Thus, it can be suggested that psychological interventions with SLSs should incorporate targeted interpersonal components that address these topics. Interventions such as interpersonal psychotherapy [55], which aims to help patients resolve interpersonal problems by employing techniques that enhance social support and reduce interpersonal stress, may be particularly effective in changing the course of coping with suicide loss [56]. Support groups for SLSs can also contribute to the healing process. They offer unique opportunities for SLSs to cope with TB, loneliness, and self-stigmatization [57] and be a place to normalize their grief experiences and share ways of coping with the suicide death [58]. Thus, it is recommended that health services proactively provide SLSs with timely and ongoing information on available support formats, including peer support groups, to minimize negative health outcomes.

More broadly, our findings suggest administering targeted postvention programs for SLSs that help them receive better social support, educating health professionals about SLSs’ psychological needs, and diminishing the stigmatization of suicide loss in the general population. These programs may aid SLSs in experiencing a greater sense of belongingness in their surroundings and, thus, enable them to cope better with the distress accompanying suicide loss [59]. It is reasonable to assume that these recommended programs, when implemented broadly and nationally, have the potential to substantially improve the life of SLSs in the aftermath of suicide loss.

## Figures and Tables

**Figure 1 ijerph-19-16561-f001:**
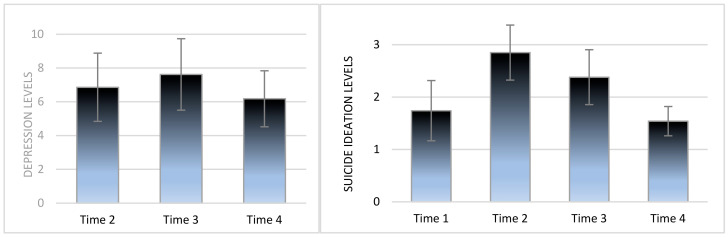
Depression and suicide ideation levels over time among suicide-loss survivors (*N* = 152).

**Figure 2 ijerph-19-16561-f002:**
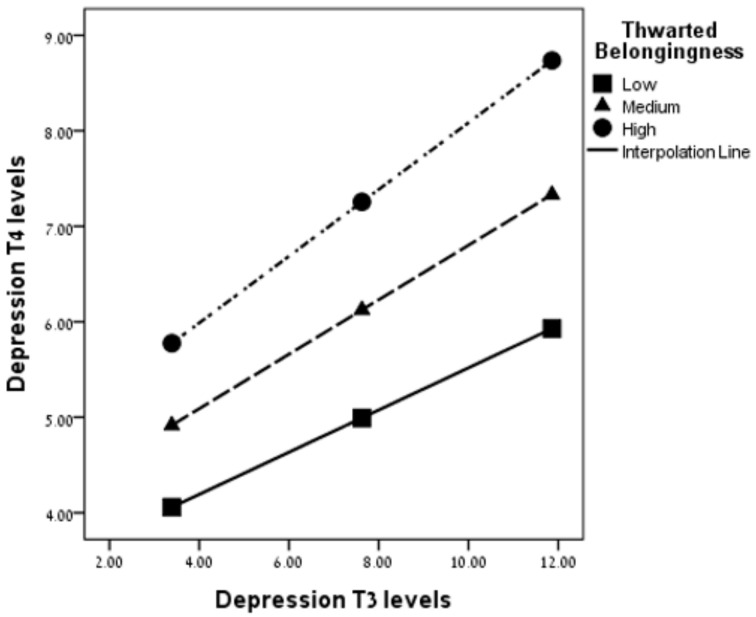
The moderation of thwarted belongingness on the association between depression-T3 and depression-T4 (*N* = 152).

**Figure 3 ijerph-19-16561-f003:**
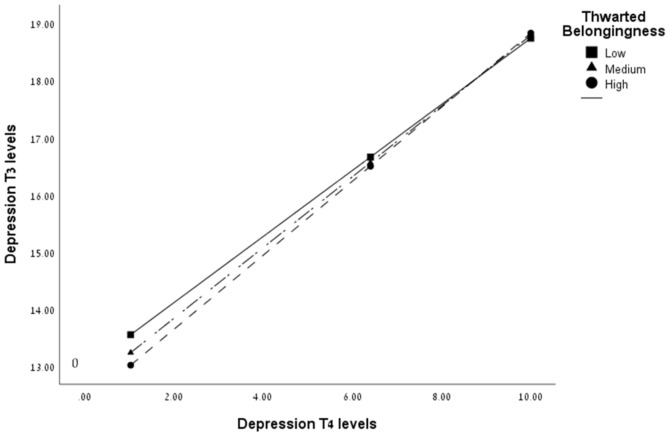
DAG representation of the moderation of thwarted belongingness on the association between depression-T4 and depression-T3 (*N* = 152).

**Table 1 ijerph-19-16561-t001:** Inter-correlations among the study variables (*N* = 152).

Variable	1	2	3	4	5	6	7	8	9	10	11	12
1. Time since suicide	–											
2. Depression-T2	−0.111	–										
3. Depression-T3	−0.055	0.320 ***	–									
4. Depression-T4	0.005	0.426 ***	0.457 ***	–								
5. SI-T1	−0.013	0.274 ***	0.168 *	0.275 ***	–							
6. SI-T2	0.115	0.173 *	0.145	0.003	0.237 **	–						
7. SI-T3	0.040	0.246 **	0.383 ***	0.356 ***	0.205 *	0.309 ***	–					
8. SI-T4	0.150	0.166 *	0.220 **	0.524 ***	0.361 ***	0.212 **	0.336 ***	–				
9. Social Support-T4	0.121	−0.076	−0.366 ***	−0.074	−0.012	−0.095	−0.107	−0.142	–			
10. Self-Disclosure-T4	0.089	−0.015	−0.200 *	−0.180 *	0.010	0.022	0.018	0.025	0.52 1 ***	–		
11. PB-T4	−0.005	0.317 ***	0.255 **	0.551 ***	0.264 ***	0.014	0.046	0.315 ***	−0.201 **	−0.242 ***	–	
12. TB-T4	−0.055	0.189 *	0.195 *	0.421 ***	0.249 **	0.130	0.124	0.318 ***	−0.490 ***	−0.428 ***	0.709 ***	–
*M*	9.72	6.86	7.62	6.18	1.74	2.85	2.38	1.54	5.47	3.66	12.9	20.35
*SD*	9.06	4.04	4.23	3.32	1.15	1.05	1.05	0.56	0.68	0.69	3.82	5.39

*Note*. * *p* < 0.05, ** *p* < 0.01, *** *p* < 0.001. Time since suicide = measured in years. Depression = as assessed by the PHQ9. SI = suicide ideation = measured by the SBQ-r Item 2. Social Support = measured by the MSPSS. Self-Disclosure = measured by the DDI. PB and TB (perceived burdensomeness and thwarted belongingness) = measured by the INQ.

**Table 2 ijerph-19-16561-t002:** Summary of Hierarchical Regression Coefficients of Depression among SLSs by Previous Depression and Interpersonal Variables. (*N* = 152).

Predictor Variables		Step 1			Step 2			Step 3			Step 4	
	B	t	β	B	t	β	B	t	β	B	t	β
Time Since Suicide	0.002	0.058	0.005	0.024	0.864	0.060	0.023	0.828	0.057	0.014	0.588	0.035
Depression-T2				2.555	4.358	0.318 ***	2.569	4.461	0.320 ***	1.653	3.130	0.206 **
Depression-T3				2.751	4.940	0.359 ***	2.971	5.072	0.387 ***	2.807	5.333	0.366 ***
Social Support T4							−1.010	−2.289	−0.191 *	−1.495	−3.484	−0.283 ***
Self-Disclosure T4							−1.052	−2.557	−0.202 *	−0.495	−1.326	−0.095
PB-T4										0.257	2.957	0.272 **
TB-T4										0.146	2.167	0.218 *
**R^2^ (∆R^2^)**	**0% (0%)**			**30% (30%)**			**33.6% (3.6%)**			**49.3% (15.7%)**		
***F* change**	***F*(1, 150) = 0.00**			***F*(2, 148) = 31.67 *****			***F*(2, 146) = 3.99 ***			***F*(2, 144) = 22.23 *****		
** *Sig.* **		**0.954**			**0.000**			**0.021**			**0.000**	

*Note*. * *p* < 0.05, ** *p* < 0.01, *** *p* < 0.001. Time since suicide = measured in years. Depression = assessed by the PHQ9. Social Support = measured by the MSPSS. Self-Disclosure = measured by the DDI. PB and TB (perceived burdensomeness and thwarted belongingness) = measured by the INQ.

**Table 3 ijerph-19-16561-t003:** Summary of Hierarchical Regression Coefficients of SI among SLSs by Previous SI and interpersonal variables (*N* = 152).

Predictor Variables		Step 1			Step 2			Step 3			Step 4	
	B	*t*	β	B	*t*	β	B	*t*	β	B	*t*	β
Time Since Suicide	0.009	1.861	0.150	0.008	1.894	0.139	0.009	2.106	0.154 *	0.009	2.083	0.148 *
SI-T1				0.145	3.969	0.300 ***	0.147	4.060	0.305 ***	0.110	2.996	0.228 **
SI-T2				0.024	0.582	0.046	0.015	0.370	0.029	0.018	0.449	0.034
SI-T3				0.135	3.305	0.255 ***	0.126	3.086	0.238 **	0.123	3.130	0.234 **
Social Support T4							−0.144	−2.086	−0.180 *	−0.094	−1.311	−0.117
Self-Disclosure T4							0.078	1.148	0.98	0.128	1.885	0.160
PB-T4										0.025	1.628	0.165
TB-T4										0.014	1.143	0.133
**R^2^ (∆R^2^)**	**2.3% (2.3%)**			**22.5% (20.2%)**			**24.7% (2.2%)**			**30.5% (5.8%)**		
***F* change**	***F*(1, 150) = 3.5**			***F*(3, 147) = 12.763 *****			***F*(2, 145) = 2.178**			***F*(2, 143) = 5.955 ******		
** *Sig.* **		**0.065**			**0.000**			**0.117**			**0.003**	

*Note*. * *p* < 0.05, ** *p* < 0.01, *** *p* < 0.001. Time since suicide = measured in years. SI = Suicide Ideation as measured by the SBQ-r item 2. Social Support = as measured by the MSPSS. Self-Disclosure = as measured by the DDI. PB & TB = Perceived Burdensomeness and Thwarted Belongingness as measured by the INQ.

## Data Availability

Due to ethical concerns, supporting data cannot be made openly available. Further information about the data and conditions for access can be received by approaching the corresponding author.

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
