# Peer review of "Depression and Suicide Ideation among Suicide-Loss Survivors: A Six-Year Longitudinal Study"

_ijerph, 2022, doi:10.3390/ijerph192416561_

Round 1

Reviewer 1 Report

Please see suggestions to improve manuscript

Author Response

Thank you very much for your valuable comments!

Attached are our answers and changes following these comments (see changes in the ms -  marked in yellow)

Reviewer 2 Report

The six-year longitudinal study was impressive.

1. Why aren't the variables measured at all times?

2. Why didn't you enter T4's suicidal ideation in 'Interpersonal effect on depression at T4'?

3. Why didn't you enter T4's depression in 'Interpersonal effect on suicide ideation at T4'?

4. Depression in Figure 1 is between 6 and 8, which is lower than expected. This needs further discussion.

5. Please describe more about the result in the discussion. (with reference to it)

Author Response

Thank you for your valuable comments! Attached are our answers (and see changes in ms - marked in yellow)
